# Clinical Remission in a 72-Year-Old Patient with a Massive Primary Cutaneous Peripheral T-Cell Lymphoma-NOS of the Eyelid, Following Combination Chemotherapy with Etoposide Plus COP

**DOI:** 10.3390/diagnostics10090629

**Published:** 2020-08-24

**Authors:** Sabina Iluta, Dragos-Alexandru Termure, Bobe Petrushev, Bogdan Fetica, Mindra-Eugenia Badea, Madalina Moldovan-Lazar, Manuela Lenghel, Csaba Csutak, Andrei Roman, Sergiu Pasca, Alina-Andreea Zimta, Ciprian Jitaru, Ciprian Tomuleasa, Rares-Calin Roman

**Affiliations:** 1Department of Hematology, Iuliu Hatieganu University of Medicine and Pharmacy, 400337 Cluj-Napoca, Romania; iluta.sabina@yahoo.com (S.I.); pasca.sergiu123@gmail.com (S.P.); 2Department of Hematology, Ion Chiricuta Clinical Cancer Center, 400124 Cluj-Napoca, Romania; Ciprianjitaru.jitaru@gmail.com; 3Department of Oral and Maxillofacial Surgery, Iuliu Hatieganu University of Medicine and Pharmacy, 400001 Cluj-Napoca, Romania; dragos.tarmure@gmail.com (D.-A.T.); madilazar@yahoo.com (M.M.-L.); rares.roman@umfcluj.ro (R.-C.R.); 4Department of Preventive Dentistry, Iuliu Hatieganu University of Medicine and Pharmacy, 400001 Cluj-Napoca, Romania; rbadea2012@gmail.com; 5Medfuture Research Center for Advanced Medicine, Iuliu Hatieganu University of Medicine and Pharmacy, 400337 Cluj-Napoca, Romania; bobe.petrushev@gmail.com (B.P.); zimta.alina.andreea@gmail.com (A.-A.Z.); 6Department of Pathology, Octavian Fodor Regional Institute of Gastroenterology and Hepatology, 400139 Cluj-Napoca, Romania; 7Department of Pathology, Ion Chiricuta Clinical Cancer Center, 400124 Cluj-Napoca, Romania; feticab@yahoo.com; 8Department of Radiology, Iuliu Hatieganu University of Medicine and Pharmacy, 400124 Cluj-Napoca, Romania; lenghel.manuela@gmail.com (M.L.); csutakcsaba@yahoo.com (C.C.); andrei.roman678@gmail.com (A.R.)

**Keywords:** primary cutaneous T-cell lymphoma, eyelid, chemotherapy, elderly patient

## Abstract

Peripheral T-cell lymphoma not otherwise specified (PTCL-NOS) is the rarest subtype of primary cutaneous lymphoma, accounting for approximately 2% of cutaneous lymphomas. The rarity of primary cutaneous PTCL-NOS means that there is a paucity of data regarding clinical and histopathological features and its clinical course. This malignancy is an aggressive and life-threatening hematological malignancy that often presents mimicking other less severe plaque-like skin conditions. Due to the nonspecific nature of these lesions, CD4-positive cutaneous T-cell lymphoma (CTCL) is often misdiagnosed as either mycosis fungoides or Sezary syndrome. We describe a patient who presented with a large tumoral mass in the right frontal area, with involvement of the right upper eyelid and the ocular globe, causing loss of vision greatly impacting the quality of life. Biopsy revealed primary cutaneous PTCL-NOS, treated successfully with cyclophosphamide, doxorubicin, vincristine, and prednisone (CHOP) plus etoposide combination chemotherapy. As elderly patients are indicated to receive attenuated doses of chemotherapy, CHOP-based regimens represent viable options.

## 1. Introduction

Peripheral T-cell lymphomas not otherwise specified (NOS) are a heterogeneous group of lymphomas that do not have well-defined criteria for being able to fit into a particular type of lymphoma [1,2,3,4]. The incidence of T-cell lymphomas in our center is small, only 8.2%. Of these, the most common is the angioimmunoblastic subtype. Primary cutaneous lymphomas are a heterogeneous group of non-Hodgkin lymphomas presenting in the skin with no evidence of lymph nodes, bone marrow, or viscera involvement at the time of diagnosis.

Cutaneous lymphomas exhibit various clinical, histological, immunophenotypic, and genetic features. Moreover, they differ in prognosis and treatment from systemic lymphomas with similar histological features [5,6,7,8]. Up to 75–80% of cutaneous lymphomas originate from T-cells, and only 20–25% from B-cells [9,10,11]. Many subtypes of cutaneous T-cell lymphomas were described, and the most common ones are mycosis fungoides and Sézary syndrome. Primary cutaneous peripheral T-cell lymphoma-NOS (PTCL-NOS) is the rarest form of cutaneous T-cell lymphomas, accounting for approximately 2% of all cutaneous lymphomas [12,13,14]. PTCL-NOS is a diagnosis of exclusion, based on the presence of typical histopathological features of lymphoma, an aberrant T-cell immunophenotype, often with a loss of CD5 and CD7, and a clonal T-cell receptor (TCR) gene rearrangement, in the appropriate clinical context [15,16].

Primary cutaneous PTCL-NOS are clinically aggressive and heterogeneous, varying from localized to generalized plaques or nodules commonly associated with constitutional symptoms [17]. Patients often present with advanced stage, lesions are often misdiagnosed, which both worsen the prognosis: five-year overall survival is about 20% [18,19]. Due to the rarity of cases, PTCL-NOS algorithms for diagnosis, workup, and treatment are lacking, especially in the elderly population. Thus, in the present manuscript we describe the successful clinical management of a PTCL-NOS of massive dimensions of the eye in a 72-year-old lady, with cardiac comorbidities.

## 2. Patient Information

A 72-year-old Romanian woman came into the Hematology service of Ion Chiricuta Clinical Cancer Center (IOCN) from Cluj-Napoca, Romania. The primary concerns as expressed by the patient were as follows: two constantly growing tumors, the loss of vision in the right eye, and the overall negative impact on her quality of life. The main symptom was itching in the tumor area, accompanied by occasional mild pain. Bleeding was noticed by the patient after low-intensity trauma to the lesion, usually caused by clothing. No spontaneous bleeding was reported. The patient was in good general health, with general B signs that included weight loss (about 15 kg in the last month) and excessive sweating. The patient had grade II hypertension but denied personal or family history of malignancy. On her initial visit, physical examination revealed a 6 cm ill-defined tumor-like lesion, red-violaceous with visible telangiectasia that presented with an ulcerated central area and crusting (Figure 1a–c). On palpation, the lesion was immobile, indurated and nontender. Upon clinical examination, we also noticed the tumefaction and skin erythema of the lower-central frontal area and glabellar area together with a 1 cm round-shaped crust. The patient did not present with palpable lymph nodes or hepato-splenomegaly. Because of the infected appearance of the lesion and the presence of odor, microbiological samples were obtained from the deepest ulcerated area and were taken to the laboratory. Microbiological analysis confirmed the clinical suspicion of infection and identified methicillin-resistant *Staphylococcus aureus* (MRSA) and *Enterobacter cloacae* complex. The patient was put on teicoplanin after the antibiogram confirmed the susceptibility of both bacterial pathogens to this antibiotic. Biologically, there was mild leukocytosis with neutrophils, and increased lactate dehydrogenase (LDH). No atypically cells were found on the blood smear. cytomegalovirus (CMV), Epstein-Barr virus (EBV), human T-cell leukemia virus (HTLV) viral investigations were performed and all came back negative.

## 3. Diagnostics

The patient reported the appearance of the first tumor two years before her first visit into our hospital. She initially presented to the doctor’s office with an infiltrated, slightly itchy, erythematous-purple plaque in the frontal level. Skin biopsy was performed in Regional County Emergency Hospital. Histopathological examination revealed polymorphous infiltrate colored mainly with the CD3, a T-cell marker. There were also numerous CD20+ B cells, which formed one or two reactive-looking lymphoid structures. The T infiltrate stained with CD4 in most cells. The Bcl-6-negativity and CD10 excluded a centrofollicular skin lymphoma. The absence of nodular CD20-positive collections did not favor the diagnosis of cutaneous lymphoma of the marginal area. The conclusion was that the clinical aspect excluded cutaneous lymphoma with small-to-medium pleomorphic T-cells, and it was in favor of a lupus erythematosus tumor. Treatment with Plaquenil (200 mg, 1 tb/day) and control over 3 months was indicated. The patient followed the treatment, but the general condition did not improve. The eyelid tumor increased, obstructing the view. The patient was then referred to a dermatology department for further investigations. The biopsy taken at the eyelid tumor level was performed again, and the biopsy was examined in a private practice pathological department. Histopathological examination revealed polymorphous inflammatory infiltrates, with predominantly small- and medium-sized lymphocytes, several granulocytes, histiocytes, and diffusely distributed plasmocytes in small nodules throughout the examined material. Immunohistochemistry analysis detected CD3-positive, CD4-positive, CD8-low positive, CD20-very low positive, CD68-positive, and CD30-negative cells. The histological aspect of the tissue sent for analysis suggested either Spiegler–Fendt sarcoid, Jessner’s lymphocytic infiltrate, or lupus erythematosus tumidus. The patient was started on hydroxychloroquine on the basis of the diagnosis, but the therapeutic choice proved to be ineffective in the following 6 months.

The symptoms did not improve, and thus the patient was referred to our department. On the basis of primary clinical evaluation, our primary diagnostic was that of mycosis fungoides (MF)/Sézary syndrome (SS). The conclusion of a final diagnosis was even more difficult because our pathologists did not have access to previous histopathological preparations. At this stage, the possibility of diagnosing MF—tumor stage having the immunophenotype of cells by their positivity for CD3, CD4, and CD7 was discussed, thus posing even more difficulty in formulating a final diagnostic. Following this, we decided to take a new biopsy from the tumor formation. The eyelid tumor was re-biopsied, and the biopsy preparation was analyzed by our team of pathologists. The hematoxylin–eosin-stained samples highlighted that the skin presented with hypodermis and ancillary structures; the tissue had a faded morphology and it was infiltrated by a malignant tumor proliferation areas, with diffuse architecture that infiltrated the dermis and subcutaneous adipose tissue, with isolated and discrete phenomena of epidermotropism, angiocentricity, and focal infiltration of skin annexes (Figure 2a,b). It was composed of atypical cells with lymphoid appearance, having a small-to-medium size with reduced cell-to-cell cohesiveness and low-to-moderate pleomorphism. The boundaries of tumor cells were hard to define and showed pale eosinophilic cytoplasm and amphophilia. The nuclei were round or oval, some of them even slightly angled. Most of the nuclei had vesicular chromatin with small nucleoli that were difficult to see, but also relatively evenly dispersed fine-grained chromatin (Figure 2c,d). Numerous mitotic areas with an increased mitotic index were also noted.

Following the immunohistochemical study, we obtained the following conclusive results. The tumor cells were intensely positive for CD3 (Figure 3b) and CD7 (Figure 3g), a relatively large number of tumor cells were positive for CD2 (Figure 3d) and CD4 (Figure 3e), and only a small population of T lymphocytes or some atypical larger cells were positive for CD8 (Figure 3f), with a CD4^+^/CD8^+^ ratio favoring CD4+ cells. Tumor cells were negative for CD30 (Figure 3i), CD5 (Figure 3c), and CD 56 (Figure 3h). A small population of atypical lymphocytes were positive in granzyme B after immunohistochemical staining. The cell proliferation index quantified by immunolabeling for Ki-67 was estimated and quantified at 80–85% (Figure 3j). Following the morphological examination correlated with special stains, the diagnosis of primary cutaneous T-cell lymphoma NOS was established. Whole-body CT was performed and revealed tumor formation in the soft epicranial parts, right frontal area, and supraorbital area, with dimensions of 22/50/45 mm. The lesion affected the upper eyelid and the paranasal region, with no signs of ocular or intracranial invasion. In the frontal on the median and parasagital line on the left, epicranial area, adjacent to the described lesion, another structure with 35/15 mm contrast socket was highlighted. The dynamic contrast-enhanced magnetic resonance imaging (MRI) revealed a large, inhomogeneous lesion, with infiltrative, invasive contour. The origin of the tumor was most probably from the periorbital superficial tissues, with invasion in the orbit and direct contact to the anteromedial aspect of the ocular globe without the presence of a cleavage plane between the two structures (Figure 4a,b and Figure 5a,b). Figure 4a shows that the T1 FSE (fast spin echo) axis and sagittal T1 FSE contrast-enhanced with fat-saturation sequence was not enhanced (a), whereas Figure 4b shows inhomogeneous enhancement of the lesion, with superficial tissue origin and invasion in the right orbit. Figure 5a,b depict the axis T2 PROPELLER sequence showing the necrotic areas in the primary and secondary lesion, satellite lesion in the superficial tissues of the frontal region (marked by arrows in the picture). A restricted diffusion on diffusion weighted imaging (DWI) sequences was observed, with an apparent diffusion coefficient (ADC) value of 0.873 × 10^−3^ mm^2^/s on the ADC map (Figure 6a). Dynamic contrast-enhanced MRI (DCE-MRI) imaging showed a type III (C) curve highly suggestive for malignancy (Figure 6b), and a high constant transfer (ktrans) value was observed, pointing to a possible good response to radiochemotherapy (Figure 7). In the proximity of the left frontal area, two nodular superficial lesions were identified, suspected of secondary nodules.

## 4. Therapy

First, in the regional hospital, where the patient was first attended, the initial therapeutic choice was a six-months regiment of hydroxychloroquine. This showed no result, since the diagnostic was incorrect, due to limited histopathological and imaging analysis of the case.

When she presented to our department, we performed a second biopsy of the tumor and gave the accurate diagnostic. Considering the age of the patient and the associated cardiac pathology, we decided to initiate combination chemotherapy using the COP plus etoposide regimen, without anthracycline. During initial evolution, we also identified that she had bacterial superinfection with MRSA in the tumor. Under appropriate antibiotic treatment, the evolution of MRSA infection was favorable. After six chemotherapy cycles, the evolution of the tumor was spectacular, with complete oncological remission of the tumoral mass. The patient regained her vision in the right eye (Figure 8a–c).

## 5. Follow-Up

The case report presented the immediate response to therapy in an impressive tumor. Unfortunately, the follow-up was not possible as the patient died of acute respiratory distress syndrome, within a month after completion of chemotherapy. This event was most probably related to her cardiovascular pathology since the lymphoma tumor was in remission. The exact cause of death was not fully established, since the pathology report was done in a local hospital.

## 6. Discussions

Over the past years, the incidence of non-Hodgkin lymphomas has steadily risen, mainly the B-cell lymphomas as they represent the great majority of non-Hodgkin’s lymphomas (NHL). T-cell and NK-cell lymphomas are generally underrepresented due to their overall low incidence, these two diseases also have a wide variability in different geographical regions and racial populations [20,21]. T- and NK-cell lymphomas are relatively rare, clinically, these subtypes are aggressive and quite heterogenous and they originate from post-thymic T-lymphocytes and NK-cells. Of all NHL, 10–15% are represented by T- and NK-cell lymphomas [22,23]. The peak incidence of T-cell lymphomas is in adults found in their sixties. These pathologies usually develop from mature T-cells and they rarely affect children. Men have a higher incidence of this disease, in comparison with women. The occurrence of adult T-cell lymphomas may be linked to previous viral infections with human T-lymphotropic virus type 1 (HTLV-1) and Epstein−Barr virus (EBV). Some medical conditions may be linked to special forms of T-cell lymphomas, like enteropathy-associated T-cell lymphoma (EATL). This form of T-cell lymphoma is associated with celiac disease and represents a rare gastrointestinal form of NHL that originates from intraepithelial T-lymphocytes [24,25,26]. Considering that, in general, T-cell lymphomas are a rarely occurring type of malignancies with a high number of subtypes, the lessons learned from each rare case are valuable. Our case report shows the importance of performing accurate and detailed histopathological and immunohistochemical characterization in case of suspecting a case of T-cell lymphoma. Our patient, after receiving a correct diagnostic of peripheral T-cell lymphoma-NOS with CD3+/CD7+/CD2+/CD7+/CD30−/CD5−/CD56− and high CD4+/CD8+ ratio, which also had MRSA infection, showed complete remission in a short period of time, after six chemotherapy cycles of COP plus etoposide regimen associated with antibiotics for MRSA.

Peripheral T-cell lymphoma not otherwise specified (PTCL-NOS) is overall the most common lymphoma type of the highly heterogenous group of T-cell lymphomas worldwide. It represents nearly 35% of all T-cell lymphomas in Europe and North America. The PTCL-NOS is a World Health Organization (WHO)-defined diagnostic category of lymphomas [27,28]. It received its name because it cannot be classified into a specific type of T-cell lymphoma. The PTCL-NOS shows association with EBV and exhibits EBV-positive neoplastic T-cells. Most commonly, older adults are affected with a slight male predominance. Generalized lymphadenopathy and B-symptoms are frequently presented, and in advanced stages, bone marrow, spleen, liver, and other tissues may be involved. Extranodal presentation is uncommon but may occur. The most frequent extranodal site is represented by the skin and gastrointestinal tract. With a five-year survival of 20–30%, the prognosis is relatively poor [4,29,30,31,32].

T-cell lymphoproliferations remain a real problem and diagnostic challenge for pathologists. Primary cutaneous T-cell lymphoma NOS is largely a diagnosis of exclusion. Among the variants and subtypes of primary cutaneous T-cell lymphoma, the following are listed: lymphoepithelioid (Lennert’s lymphoma): numerous histiocytes clustered along with T-cell; T-cells, commonly scattered, larger, more atypical, and pleomorphic cells, including occasional Reed−Sternberg-like cells (usually EBV+); in these, the T-cell subtype, often CD8+. The T-zone lymphoma is no longer considered a variant of PTCL-NOS. Instead, it is considered to have a nonspecific morphologic pattern, which can be seen in PTCL-NOS cases with helper T-cell phenotype that are reclassified as angioimmunoblastic T-cell lymphoma (AITL) or other nodal lymphomas of T follicular helper cell origin. The other excluded variant is the follicular variant that is also no longer considered a variant of PTCL-NOS and was reclassified by WHO as AITL and other nodal lymphomas of T follicular helper cell origin. In peripheral T-cell lymphoma associated with B-cell proliferation, the B-cells are small, alongside mature plasma cells and plasmacytoid large B lymphocytes or plasmablastic B-cells. These are often EBV+.

The differential diagnosis of primary cutaneous T-cell lymphoma NOS includes the following entities: angioimmunoblastic T-cell lymphoma (AITL) which has similar morphology to NOS; but it may be more prominent in arborizing high endothelial venules in AITL. The infiltrate may be more polymorphous than in PTCL-NOS. B-cells EBV+ are present in most cases of AITL. The primary cutaneous presentation being extreme; anaplastic large-cell lymphoma, ALK-positive and ALK-negative (ALCL), has cells with hallmark morphology, sinus involvement, and anaplastic cytologic features supporting ALCL. The ALCL is strong and has uniform CD30+ cells in a membranous and paranuclear pattern. CD30 marker can also be focally positive and represented in a lower proportion in PTCL-NOS. ALK-negative subtype shows no evidence of ALK locus abnormalities; whereas the ALK-positive subtype presents with ALK locus abnormalities, more precisely(2;5)(p23;q35) NPM1/ALK. Histologically and immunophenotypically, PTCL and adult T-cell leukemia/lymphoma (ATLL) can be indistinguishable. However, adult T-cell leukemia/lymphoma is caused by infection by human T-cell leukemia virus type 1 (HTLV-1). HTLV-1 is integrated into the genome of neoplastic cells. Patients with ATLL often present with hypercalcemia. The leukemic phase is commonly encountered in ATLL patients. Multilobate and flower-shaped cells are in blood smear. In subcutaneous panniculitis-like T-cell lymphoma, the patients present with subcutaneous nodules, usually without other signs of disease. Atypical lymphoid infiltrates in subcutaneous tissue that involves fat lobules, typically spares septa, overlying dermis, and epidermis, and the tumor cells are CD3+ and CD8+. In typical cases of the nasal type of extranodal NK-/T-Cell lymphoma, the patients present only with extranodal disease. Rare cases have been described at the upper aerodigestive tract level and in extranasal sites: skin, soft tissue, lower gastrointestinal, and testes. This entity is constituted by polymorphous lymphoid infiltrate of variable morphology and/or angioinvasion, often associated with extensive fibrinoid necrosis. The tumor cells are CD3+/−, CD2+, CD56+, and EBV+/−. Skin involvement of enteropathy-associated T-cell lymphoma and hepatosplenic T-cell lymphoma is unusual, and, in our case, they were clinically excluded.

When it comes to interpreting immunohistochemistry (IHC) biomarkers staining for the diagnostic of PTCL-NOS, it is important to include as many as possible since these can only be interpreted as a cluster. For instance, CD30 can either be absent or slightly positive, while CD56 can rarely be positive. CD68 expression with high positivity of CD8 is typical for indolent acral CD8+ T-cell lymphoma that is usually a benign tumor. CD20 can be either less than 1% positive or entirely negative. CD3+/CD4+/CD8− shows a less aggressive type of PTCL-NOS, as compared with CD8 strongly positive PTCL-NOS. An aggressive type of PTCL-NOS is also characterized by positivity for granzyme B [33,34,35,36].

A more detailed description of each immunohistochemical marker that we used in our study and how it helped us differentiate between different types of cutaneous lymphomas is found in Table 1.

## 7. Conclusions

As elderly patients are indicated to receive attenuated doses of chemotherapy, CHOP-based regimens represent viable options [4,5]. Still, considering that our patient had cardiac comorbidities, we chose to remove anthracyclines from the combination treatment and replaced it with etoposide. The particularity of this case is the impressive huge dimensions of the tumoral mass of the eye. Moreover, the impressive feature of this case is that the patient even regained her vision, proving thus that etoposide plus COP is a viable alternative for such a clinical scenario: elderly patients with invasion of internal organs.

## Figures and Tables

**Figure 1 diagnostics-10-00629-f001:**
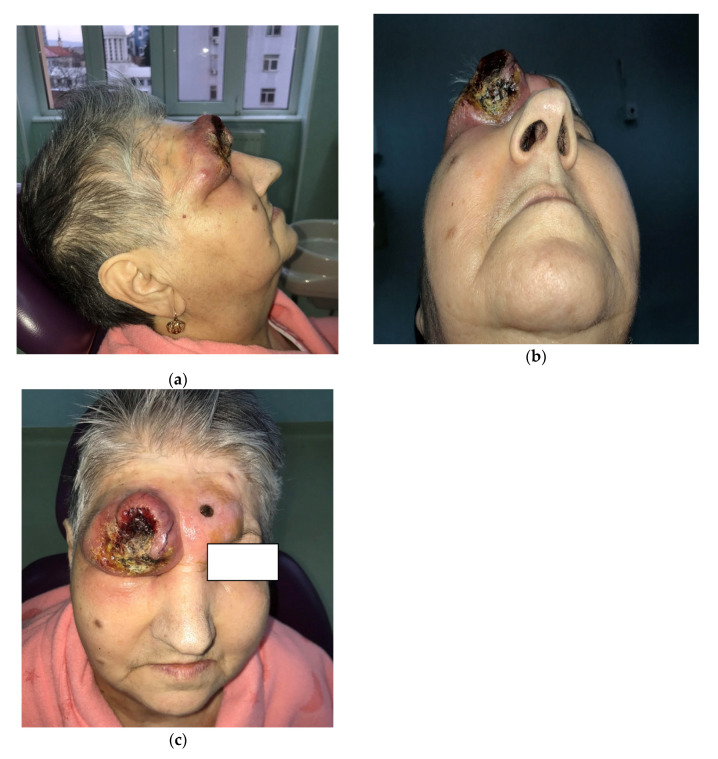
(**a**–**c**) The initial visit, which revealed a 6 cm ill-defined tumor-like lesion, red-violaceous with visible telangiectasia that presented with an ulcerated central area and crusting.

**Figure 2 diagnostics-10-00629-f002:**
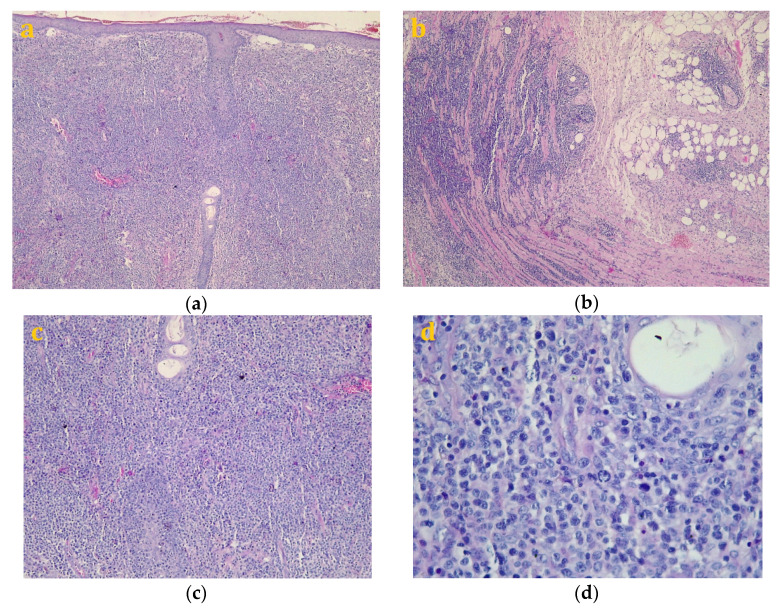
Pathology diagnosis of a primary cutaneous peripheral T-cell lymphoma not otherwise specified—Hematoxylin–Eosin staining. ((**a**), Magnification 40×, (**b**), magnification 100×, (**c**), Magnification 200×, (**d**), Magnification 400×).

**Figure 3 diagnostics-10-00629-f003:**
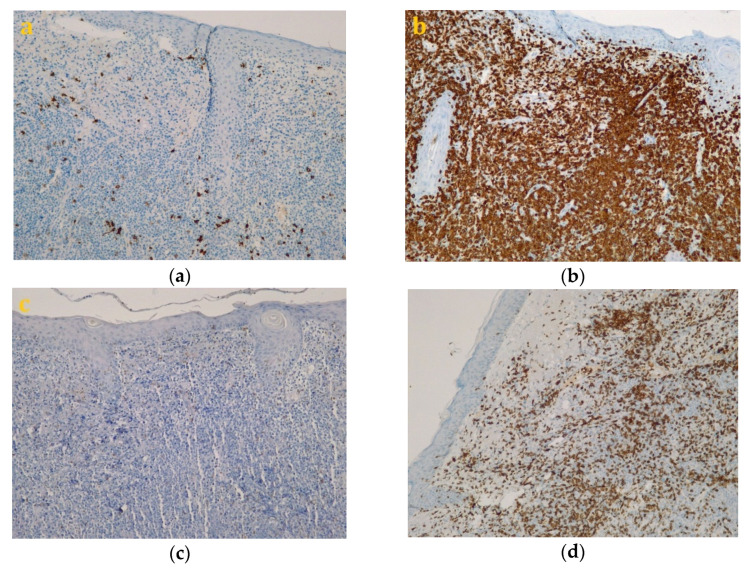
Immunohistochemistry staining for the following markers: CD20 (**a**, Magnification 200×), CD3 (**b**, Magnification 200×), CD5 (**c**, Magnification 200×), CD2 (**d**, Magnification 200×), CD4 (**e**, Magnification 200×), CD8 (**f**, Magnification 200×), CD7 (**g**, Magnification 200×), CD56 (**h**, Magnification 200×), CD30 (**i**, Magnification 200×), Ki-67 (**j**, Magnification 200×).

**Figure 4 diagnostics-10-00629-f004:**
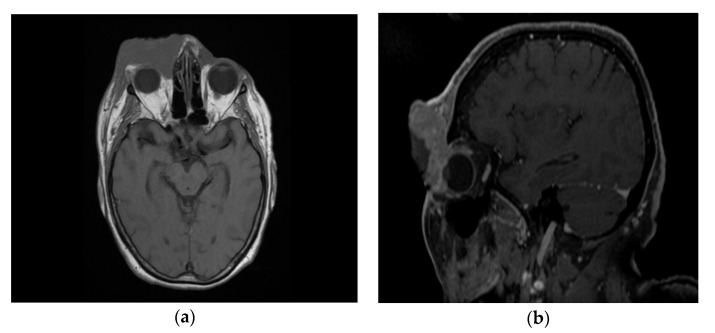
Ax T1 FSE non-enhanced (**a**) and sagittal T1 FSE contrast-enhanced with fat-saturation sequence (**b**). Inhomogeneous enhancement of the lesion, with superficial tissue origin and invasion in the right orbit.

**Figure 5 diagnostics-10-00629-f005:**
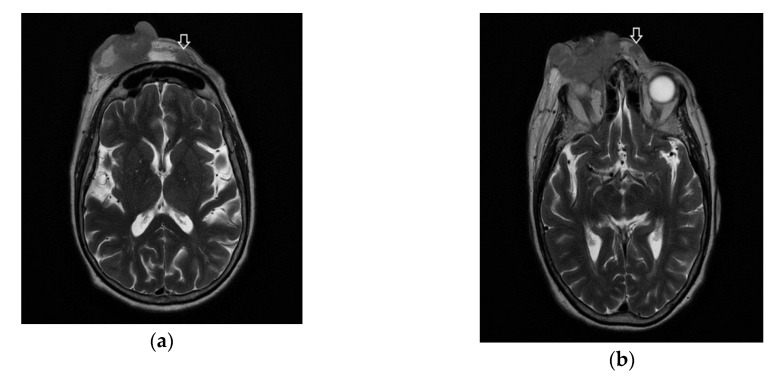
(**a**,**b**) The ax T2 PROPELLER sequence showing the necrotic areas in the primary lesion and secondary, satellite lesion in the superficial tissues of the frontal region (arrows).

**Figure 6 diagnostics-10-00629-f006:**
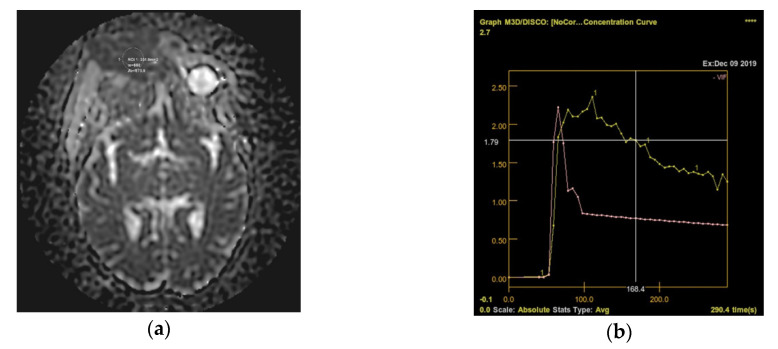
(**a**) Restricted diffusion on DWI sequences was observed, with an ADC value of 0.873 × 10^−3^ mm^2^/s on the ADC map; (**b**) Dynamic contrast-enhanced MRI (DCE-MRI) imaging showed a type III (C) curve highly suggestive for malignancy.

**Figure 7 diagnostics-10-00629-f007:**
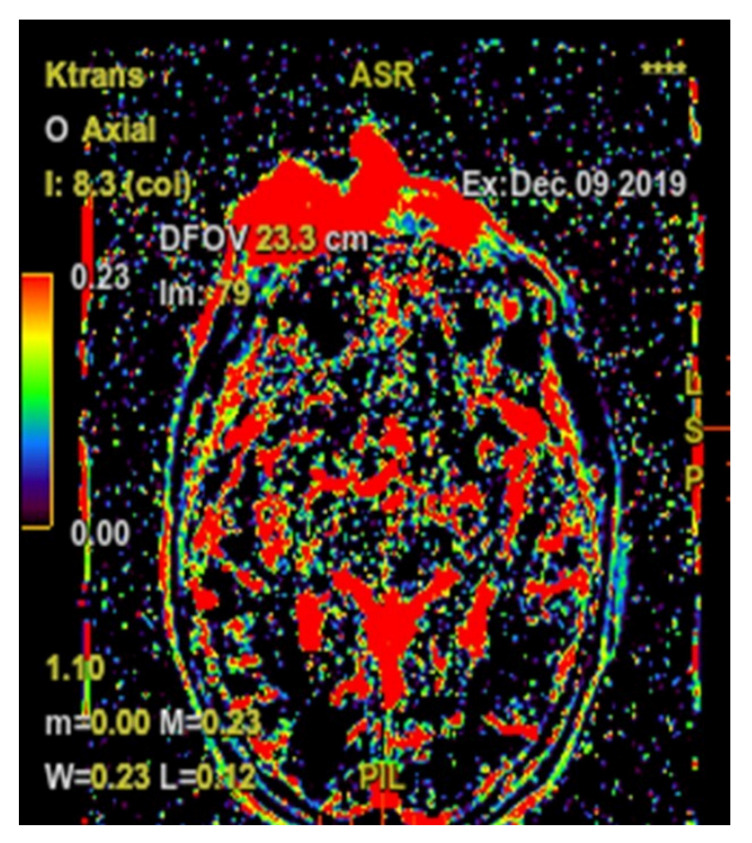
High constant transfer (ktrans) value was observed, pointing to a possible good response for radiochemotherapy.

**Figure 8 diagnostics-10-00629-f008:**
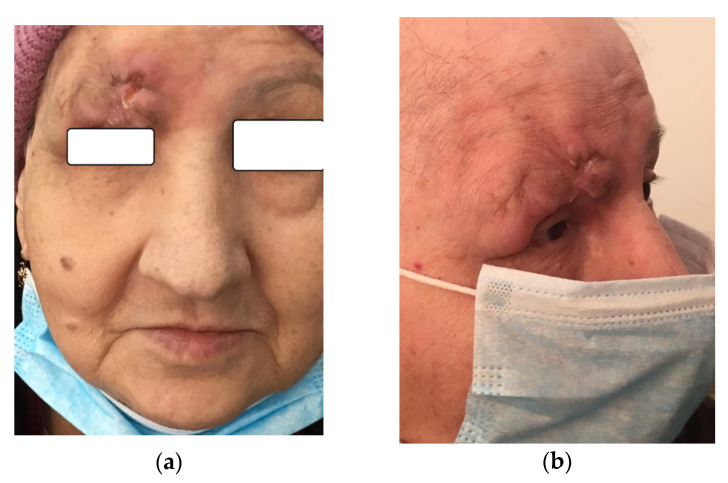
(**a**–**c**) Clinical remission following chemotherapy.

**Table 1 diagnostics-10-00629-t001:** Track-record of the pathology diagnosis

Round	Marker	Status	Meaning
2nd Biopsy	CD3	Positive	A marker typical for T-cell, used to validate T-cell origin of the tumor
	CD4	Positive	Specific with peripheral T-cell lymphoma, not otherwise specified (PTL-NOS), which is also commonly associated with loss of other T-cell markers
	CD8	Positive	Excluding Reed−Sternberg-like cells (usually EBV+) diagnostic and subcutaneous panniculitis-like T-cell, high positivity of CD8 shows a more aggressive type of PTCL Lymphoma, which has a higher population of CD8+ cells
	CD20	A very low proportion of cells	CD20+ should be less than 1% of stained cells
	CD68	Positive	Specific with PTL-NOS, however little information is known about this marker
	CD30	Negative	Usually negative in PTCL-NOS
3rd biopsy	CD3	Positive	A marker typical for T-cell, used to validate T-cell origin of the tumor
	CD7	Positive	Typical T-cell marker, PTCL-NOS commonly express it
	CD2	High number of positive cells	Typical T-cell marker, PTCL-NOS commonly express it
	CD4	High number of positive cells	High positivity of CD4 shows a less aggressive type of PTCL lymphoma
	CD8	Small population of positive cells	excluding Reed−Sternberg-like cells (usually EBV+) diagnostic and subcutaneous panniculitis-like T-cell, high positivity of CD8 shows a more aggressive type of PTCL lymphoma, which has a higher population of CD8+ cells
	CD30	Negative	Exclusion of anaplastic large-cell lymphoma ALK-positive and ALK-negative (ALCL), which is strongly CD30+
	CD5	Negative	Loss of CD5 is specific for PTCL-NOS
	CD56	Negative	Exclusion of extranodal NK-/T-Cell lymphoma, which is CD56+
	Granzyme B	Small population of positive cells	Specific for cytotoxic T-cell lymphomas, a more aggressive phenotype
	Ki-67	Proliferation index of 80–85%	Verify the proliferation activity of cells

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
