# Peer review of "Clinical Remission in a 72-Year-Old Patient with a Massive Primary Cutaneous Peripheral T-Cell Lymphoma-NOS of the Eyelid, Following Combination Chemotherapy with Etoposide Plus COP"

_diagnostics, 2020, doi:10.3390/diagnostics10090629_

Round 1

Reviewer 1 Report

Your manuscript provides a richly detailed description of a very fascinating case. However, revision by an English native speaker is advised. Writing should be more concise.

Past tense is generally more suitable when describing a case. Avoid using present tense in one sentence and past tense in the following one.

The cause of death should be specified (e.g. what triggered the ARDS), if possible/known.

In the Discussion section, you offer an overview on NHLs, however perhaps cutaneous lymphomas and particularly PCTCL-NOS would merit more space.

Corrections:

Change "The patient has a history grade II additional high-risk primary hypertension. She has no personal or family history of malignancy." with "The patient had grade II hypertension, but denied personal or family history of malignancy".

Change "The patient din does not present with palpable lymph nodes or hepato-splenomegaly" with "The patient did not present with palpable lymph nodes or hepato-splenomegaly"

Change "Microbiological analyses conformed lesion infection and the bacteria identified was methicillin-resistant Staphylococcus aureus (MRSA) and Enterobacter cloacae complex." with "Microbiological analysis confirmed the clinical suspicion of infection and identified methicillin-resistant Staphylococcus aureus (MRSA) and Enterobacter cloacae complex."

Change "The cell negativity for Bcl-6" with "Bcl-6 negativity"

Clarify "contrast socket": did you mean peripheral contrast uptake?

Author Response

Dear editor,

              Thank you very much for reviewing our manuscript. We appreciate the tremendous effort and time the reviewers devoted to improving our manuscript. We sincerely feel that their thoughtful comments have further strengthened the manuscript. Specific responses to each comment are presented in the Responses to the Reviewers. In the revised manuscript, revisions to the manuscript are indicated in red font. We hope that our responses to the reviewers’ comments and the revisions made to the manuscript satisfy all questions and concerns. 

              With my best regards,

            Ciprian Tomuleasa, M.D.

             Department of Hematology,

            Iuliu Hatieganu University of Medicine and Pharmacy, Cluj Napoca, Romania.

Comments to the Reviewers

Reviewer #1:

provides a richly detailed description of a very fascinating case. However, revision by an English native speaker is advised. Writing should be more concise.

The text was carefully checked for English grammar by a third party. Where necessary the long phrases were reduced to shorter sentences in order to make the text more concise. The off phrasing or misuse of word was replaced or excluded. We are confident that in the new form, our case report in more complete and easier to understand.

Past tense is generally more suitable when describing a case. Avoid using present tense in one sentence and past tense in the following one.

We replaced all verbs from case description to the past tense form. Thank you.

The cause of death should be specified (e.g. what triggered the ARDS), if possible/known.

We are not fully certain on the cause of death. The suspected cause of death was most probably related to the patients’ cardiovascular comorbidities. We mentioned the following in the text: “This event was most probably related to her cardiovascular pathology, since the lymphoma tumor was in remission. The exact cause of death was not fully established, since the pathology report was done in a local hospital.”

In the Discussion section, you offer an overview on NHLs, however perhaps cutaneous lymphomas and particularly PCTCL-NOS would merit more space.

We described in more detail in the discussion section about different types of NHLs, in particular PCTCL-NOS. Thank you.

Corrections:

Change "The patient has a history grade II additional high-risk primary hypertension. She has no personal or family history of malignancy." with "The patient had grade II hypertension, but denied personal or family history of malignancy".

We made the correction. Thank you.

Change "The patient din does not present with palpable lymph nodes or hepato-splenomegaly" with "The patient did not present with palpable lymph nodes or hepato-splenomegaly"

We made the correction. Thank you.

Change "Microbiological analyses conformed lesion infection and the bacteria identified was methicillin-resistant Staphylococcus aureus (MRSA) and Enterobacter cloacae complex." with "Microbiological analysis confirmed the clinical suspicion of infection and identified methicillin-resistant Staphylococcus aureus (MRSA) and Enterobacter cloacae complex."

We made the correction. Thank you.

Change "The cell negativity for Bcl-6" with "Bcl-6 negativity"

We made the correction. Thank you.

Clarify "contrast socket": did you mean peripheral contrast uptake?

We made the correction. Thank you.

Reviewer 2 Report

Overall, this is a well described report of a rare presentation of periorbital lymphoma.  It would be useful to most clinicians if the authors could comment further on the delay in diagnosis and the challenges of interpreting the immunohistochemical markers.  Creating a chart of the immunohistochemistry for each of the biopsies with a brief description of the importance of positivity or negativity for each the major marker would be helpful. 

Author Response

Dear editor,

              Thank you very much for reviewing our manuscript. We appreciate the tremendous effort and time the reviewers devoted to improving our manuscript. We sincerely feel that their thoughtful comments have further strengthened the manuscript. Specific responses to each comment are presented in the Responses to the Reviewers. In the revised manuscript, revisions to the manuscript are indicated in red font. We hope that our responses to the reviewers’ comments and the revisions made to the manuscript satisfy all questions and concerns. 

              With my best regards,

            Ciprian Tomuleasa, M.D.

             Department of Hematology,

            Iuliu Hatieganu University of Medicine and Pharmacy, Cluj Napoca, Romania.

Comments to the Reviewers

Reviewer #2:

Overall, this is a well described report of a rare presentation of periorbital lymphoma.  It would be useful to most clinicians if the authors could comment further on the delay in diagnosis and the challenges of interpreting the immunohistochemical markers.  Creating a chart of the immunohistochemistry for each of the biopsies with a brief description of the importance of positivity or negativity for each the major marker would be helpful.

Thank you for the suggestion, which in deep increased the value of our case report. We included in the discussion section more details about the diagnostic of T-cell lymphoma and included a paragraph describing the flow of analyzing the immunohistochemical markers.

We also provided at the send of discussion section a table showing the diagnostic importance of each biomarker and the IHC results from each of the two rounds of biopsy.
